# INTERPRETING QUANTUM CIRCUIT LEARNING WITH QPERT: A STEP TOWARD TRUSTWORTHY QUANTUM AI

## ABSTRACT

Quantum Circuit Learning (QCL) presents a promising hybrid computational framework that combines the representational capacity of parameterized quantum circuits (PQCs) with classical optimization techniques for solving machine learning problems. However, the opaque nature of QCL models limits their adoption in domains requiring transparency and accountability. In this work, we introduce quantum perturbation (QPERT), a novel perturbation-based explainability approach tailored for QCL. QPERT generates a saliency mask by quantifying the importance of input features for a given instance while preserving key quantum properties such as entanglement and superposition. We evaluate QPERT in explaining a hybrid quantum-classical architecture trained on the Iris dataset. Comparative analysis against established explainability techniques, including SHAP and LIME, highlights QPERT's effectiveness in delivering interpretable insights into quantum model behavior. Our results demonstrate the feasibility of interpretable quantum learning and offer practical guidance for integrating explainability into quantum-classical pipelines.

## 1 INTRODUCTION

Quantum Machine Learning (QML) has emerged as a rapidly evolving interdisciplinary domain at the intersection of quantum computing and artificial intelligence, with the aim of exploiting uniquely quantum mechanical phenomena, notably superposition and entanglement, to achieve computational capabilities beyond those attainable with classical architectures Zeguendry et al. (2023). Recent developments, both theoretical and experimental, indicate that quantum algorithms can yield polynomial or even exponential performance enhancements for certain classes of machine learning problems, particularly those involving high-dimensional feature representations or optimization landscapes that are classically intractable Biamonte et al. (2017); Benedetti et al. (2019); Ghosh & Ghosh (2024).

Within this landscape, *Quantum Circuit Learning* has garnered significant attention as a practical and versatile paradigm for near-term quantum Zhai (2022). QCL leverages parameterized quantum circuits (PQCs) as trainable models, wherein adjustable quantum gates embed classical data into quantum states, execute quantum transformations, and generate predictive outputs via projective measurements Li et al. (2024). This hybrid quantum–classical framework integrates naturally with gradient-based optimization techniques prevalent in contemporary machine learning, thereby enabling efficient parameter tuning while potentially harnessing intrinsic quantum computational advantages Mitarai et al. (2018).

However, the deployment of QCL in high-stakes domains such as scientific research, healthcare, finance, and autonomous systems faces a critical barrier: the lack of interpretability Gil-Fuster et al. (2024). Quantum measurements introduce fundamental stochasticity, entangled states create non-local feature correlations, and the probabilistic nature of quantum outputs conflicts with the deterministic assumptions underlying classical explainable AI (XAI) methods such as SHAP Lundberg & Lee (2017) and LIME Ribeiro et al. (2016). For instance, when SHAP attempts to compute Shapley values for a quantum classifier, the measurement-induced randomness can lead to inconsistent feature attributions across identical inputs, undermining explanation reliability Heese et al. (2025).

This interpretability gap is particularly problematic because recent studies demonstrate that many existing XAI methods exhibit high sensitivity to input perturbations, producing dramatically different explanations for inputs that yield identical model predictions Alvarez-Melis & Jaakkola (2018). In quantum systems, this instability is amplified by measurement noise and quantum decoherence, making robust explanation generation an even more pressing concern. Furthermore, naive application of classical perturbation-based methods can inadvertently destroy quantum correlations encoded in the input representation, leading to explanations that reflect classical shadows of quantum computations rather than genuine quantum feature importance.

This work addresses the challenge of interpretability in quantum circuit learning by introducing QPERT, a model-agnostic, perturbation-based explanation method. QPERT employs a gradient-driven optimization strategy to systematically evaluate the impact of perturbing individual input features on the predictions of QCL models. Through this process, it generates a saliency mask that highlights the relative importance of each input dimension. To ensure the reliability and plausibility of the explanations, QPERT incorporates regularization mechanisms that constrain perturbations to remain within the distribution of inputs encountered during training, thereby preserving the semantic and statistical coherence of the perturbed instances.

Our key contributions in this work are as follows:

- We introduce **QPERT**, the first learning-based perturbation framework designed specifically for QCL, which captures and preserves quantum-specific properties such as entanglement and superposition during explanation generation.
- We conduct comprehensive experiments on a widely used, publicly available dataset Iris using our proposed QCL architecture and demonstrate that QPERT consistently outperforms state-of-the-art explainability methods, including SHAP and LIME, in generating more faithful and informative explanations.

## 2 RELATED WORK

The field of explainable artificial intelligence has developed numerous approaches to interpret complex machine learning models. Model-agnostic methods such as SHAP Lundberg & Lee (2017) and LIME Ribeiro et al. (2016); Kashyap et al. (2025) have gained widespread adoption due to their general applicability and theoretical foundations. SHAP attributes prediction contributions to input features using Shapley values from cooperative game theory den Broeck et al. (2020), providing a unified framework that satisfies desirable axioms including efficiency, symmetry, and additivity. LIME creates local linear approximations around individual predictions by fitting interpretable models to perturbed inputs in the neighborhood of the instance being explained Chowdhury et al. (2022).

The application of classical XAI methods to quantum machine learning models has revealed fundamental compatibility issues. Quantum measurements introduce inherent randomness Coleman et al. (2020) that conflicts with the deterministic assumptions underlying most classical explanation techniques Barua et al. (2025). When a quantum circuit is measured repeatedly with identical inputs, the resulting outputs exhibit quantum-statistical variation that classical methods interpret as model uncertainty rather than fundamental quantum behavior. Several recent works have attempted to adapt classical XAI frameworks to quantum settings Barua et al. (2025). qSHAP Steinmüller et al. (2022) modifies the traditional SHAP approach by incorporating Fourier analysis to handle the periodic structure of parameterized quantum circuits. This method accounts for the fact that quantum gate parameters exhibit $2\pi$-periodicity, which standard SHAP implementations fail to capture. Q-LIME $\pi$ Vargas (2024) introduces quantum-inspired perturbation schemes that attempt to preserve quantum coherence properties during local explanation generation.

Perturbation-based attribution methods form another important category, directly measuring feature importance by observing prediction changes under input modifications. The PERT framework Parvatharaju et al. (2021) demonstrates sophisticated perturbation-based explanation for time series classification. However, direct application of PERT to quantum systems is non-trivial due to fundamental differences between temporal correlations in classical time series and quantum correlations in QCL inputs. Quantum correlations can be non-local and exhibit interference effects that have no classical analog, requiring specialized perturbation strategies that preserve quantum-relevant statistical properties.

However, recent research has revealed significant limitations in the robustness of these classical XAI methods. Alvarez-Melis and Jaakkola Alvarez-Melis & Jaakkola (2018) demonstrated that explanation methods exhibit high sensitivity to small input perturbations, producing inconsistent attributions even when model predictions remain stable. This instability is particularly pronounced in perturbation-based methods, which rely on sampling strategies that may not adequately capture the underlying data manifold. In parallel with classical methods, quantum-native approaches have emerged that focus on architectural interpretability through gate-level analysis Buonaiuto et al. (2024); Pira & Ferrie (2024). SVQXs (Shapley Values for Quantum Explanations) Heese et al. (2025) assign importance scores to individual quantum gates based on metrics such as expressibility, entanglement capability, and hardware fidelity. This method provides insights into which components of a quantum circuit contribute most significantly to model performance, but operates at the circuit architecture level rather than input feature level.

Despite these advances, significant gaps remain in quantum machine learning interpretability. Existing quantum XAI methods require strong assumptions about the quantum system's structure or operation, limiting their applicability to diverse QCL architectures. Most critically, no existing method adequately addresses the fundamental tension between perturbation-based explanation and quantum correlation preservation. Classical perturbation methods treat input features as independent variables, potentially destroying entangled or superposed relationships that are central to quantum computational advantage. This limitation is particularly problematic for QCL models where quantum correlations between input features may be essential for model performance. Our work addresses these limitations by introducing a perturbation-based framework that explicitly preserves quantum-relevant input properties while generating stable, faithful explanations for diverse QCL architectures.

## 3 METHODOLOGY

To identify which input features influence QCL predictions, we introduce a saliency-based optimization architecture (Figure 1) that perturbs input instances using a learnable mask. Each mask entry controls the interpolation between the original input and background samples drawn from the dataset, enabling localized, in-distribution perturbations. The perturbed instance during prediction is re-encoded into a quantum state and passed through the QCL model, yielding a prediction whose divergence from the original output is measured. A composite loss function, which includes target prediction suppression, sparsity (L1 regularization), and quantum consistency terms such as fidelity, entanglement, and superposition, is minimized over multiple iterations using gradient descent. The mask is updated through backpropagation and its final values are normalized to produce saliency scores, highlighting the features that affect the model's quantum decision process the most.

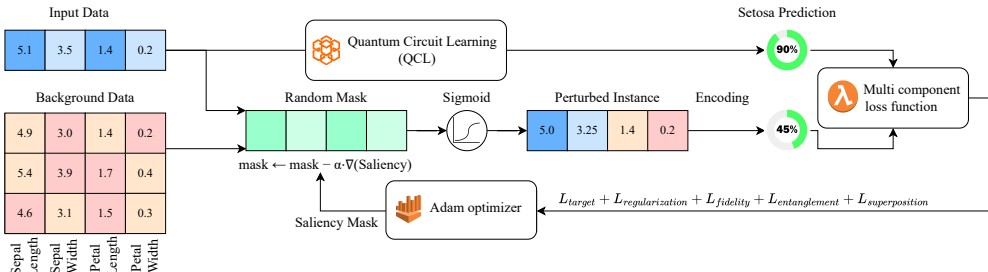

Figure 1: QPERT Architecture

We construct a QCL model consisting of a data-encoding layer, a parameterized variational block, and a final measurement layer. The model uses the full 4-qubit Hilbert space instead of discarding information, ensuring that all qubits contribute toward classification. This design enables the representation of all 16 basis states $(2^4)$, which are mapped to 3 output classes through a probabilistic decoding scheme. To enhance expressivity, output rotations are applied to each qubit prior to measurement. Gate parameters are optimized using classical methods, such as gradient descent with the parameter-shift rule and learning rate scheduling. To apply post hoc explanation techniques such as

SHAP and LIME, we generated perturbed input samples and evaluated the output probabilities of the QCL model. Due to inherent quantum noise and stochasticity, we average multiple measurement shots per inference. The sampling strategy and model-query interface are adapted to accommodate quantum-specific constraints, including entanglement dependencies and shot variance.

**Quantum-Inspired Loss Functions:** To stabilize interpretability, we define custom regularization losses, made up of three components:

**1. Fidelity Loss**: The fidelity loss assesses the impact of perturbations on the representation of the quantum state of the input data. Quantum fidelity $F$ measures the similarity between two quantum states, ranging from 0 (orthogonal states) to 1 (identical states), effectively quantifying their closeness Muller (2023). By minimizing $1 - F$, we penalize perturbations that drastically alter the quantum state, encouraging explanations that preserve the essential quantum information content.

$$L_{\text{fidelity}} = 1 - F(\psi_{\text{original}}, \psi_{\text{perturbed}}) \tag{1}$$

where the quantum fidelity is defined as:

$$F(\psi_1, \psi_2) = |\langle\psi_1|\psi_2\rangle|^2 \tag{2}$$

When direct quantum state access is unavailable, the loss returns to Jensen-Shannon divergence Hoyos-Osorio & Sanchez-Giraldo (2024); Majtey et al. (2005) between prediction probability distributions, which serves as a classical proxy for quantum fidelity and maintains the same interpretability to penalize dramatic state changes.

$$L_{\text{fidelity}} = JS(P_{\text{original}}, P_{\text{perturbed}}) \tag{3}$$

where the Jensen–Shannon Divergence is defined as

$$JS(P \parallel Q) = \frac{1}{2}D_{\text{KL}}(P \parallel M) + \frac{1}{2}D_{\text{KL}}(Q \parallel M), \quad M = \frac{1}{2}(P + Q) \tag{4}$$

and equivalently expressed in terms of Shannon entropy $H(\cdot)$ as

$$JS(P \parallel Q) = H\left(\frac{P+Q}{2}\right) - \frac{1}{2}H(P) - \frac{1}{2}H(Q) \tag{5}$$

where the Shannon entropy is defined as

$$H(P) = -\sum_i P(i) \log P(i) \tag{6}$$

**2. Entanglement Loss**: The entanglement loss is computed from the full Pearson correlation matrix Jebarathinam et al. (2020) of the perturbation mask elements and penalizes non-zero correlations between distinct mask entries. Intuitively, we form the correlation matrix of mask elements, extract the off-diagonal correlations which represent pairwise dependencies between different features/elements, take their absolute values, and minimize their mean. Under this formulation, reducing the loss suppresses spurious linear correlations between mask elements and thus controls the degree of classical correlation retained in the perturbation mask.

Let $m \in \mathbb{R}^P$ denote the flattened perturbation mask with elements $m_p$. Define the Pearson correlation matrix $R \in \mathbb{R}^{P \times P}$ with entries

$$R_{pq} = \rho_{pq} = \frac{\mathbb{E}[(m_p - \bar{m}_p)(m_q - \bar{m}_q)]}{\sqrt{\mathbb{E}[(m_p - \bar{m}_p)^2]} \sqrt{\mathbb{E}[(m_q - \bar{m}_q)^2]}} \tag{7}$$

where $\bar{m}_p$ is the mean of element $p$ over the chosen axis, e.g., batch or time.

Extract the off-diagonal set $\mathcal{O} = \{(p, q) \mid p \neq q\}$, and compute the entanglement loss as:

$$\mathcal{L}_{\text{entanglement}} = \frac{1}{|\mathcal{O}|} \sum_{(p,q) \in \mathcal{O}} |R_{pq}| \tag{8}$$

Minimizing $\mathcal{L}_{\text{entanglement}}$ reduces linear dependencies between distinct mask entries, thereby controlling unwanted correlations.

**3. Superposition Loss**: The superposition loss penalizes sparse probability distributions in the model predictions and, as a result, prevents the collapse of quantum states. Quantum superposition allows particles to exist in multiple states simultaneously until measurement causes collapse to a definite state Daley et al. (2022). This loss function promotes distributional diversity by integrating two complementary metrics: participation ratio, which assesses the number of states contributing significantly to the superposition, and Shannon entropy, which quantifies the distribution's uniformity. The participation ratio penalizes peaked distributions approaching classical definiteness, whereas the Shannon entropy term rewards uncertainty across multiple outcomes. By combining these components, the loss function ensures that perturbations preserve the quantum-like capacity for representing multiple possibilities simultaneously, preventing premature collapse to deterministic classical states.

$$\mathcal{L}_{\text{superposition}} = \mathcal{L}_{\text{participation}} + \alpha \cdot \mathcal{L}_{\text{entropy}} \tag{9}$$

$$\mathcal{L}_{\text{participation}} = \frac{\sum_i p_i^2 - \frac{1}{N}}{1 - \frac{1}{N}} \tag{10}$$

$$\mathcal{L}_{\text{entropy}} = 1 - \frac{H(p)}{\log N} = 1 - \frac{-\sum_i p_i \log p_i}{\log N} \tag{11}$$

where $p_i$ are the predicted class probabilities, $\alpha$ is the scaling parameter , $N$ is the number of classes, and $H(p)$ is the Shannon entropy.

## 4 RESULTS

To evaluate the interpretability and quantum-consistency of QPERT, we present empirical results across local and global explanations, direct saliency analysis, and the training dynamics of quantum-inspired loss components. The analysis is conducted on a QCL model trained to classify Iris species.

### 4.1 LOCAL EXPLANATION WITH LIME

Figure 2 illustrates a LIME-based explanation for a test instance classified as *Virginica* with a confidence of 0.51. The most influential features were low petal length and petal width, contributing significantly to the model's decision. This aligns with botanical intuition Virginica is characterized by longer and wider petals, and, relatively longer sepals, demonstrating that QCL is capable of internalizing semantically meaningful decision boundaries. Sepal width had negligible influence, reinforcing that the QCL model's decision is dominated by class-specific morphological attributes.

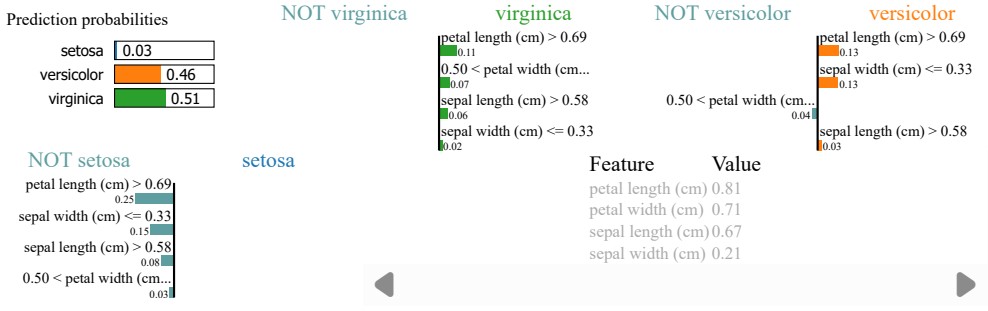

Figure 2: LIME explanation indicating key contributing features for classifying a sample as Virginica.

## 4.2 GLOBAL EXPLANATION WITH SHAP

Figure 3 presents SHAP summary plots for each class. For Setosa, high SHAP values correspond to low petal length and width, with petal length being the most impactful feature. For Versicolor, the model relies more on higher values of petal width and length. These global explanations suggest the QCL model not only learns to differentiate between classes but also aligns feature attributions with the biological structure of the dataset.

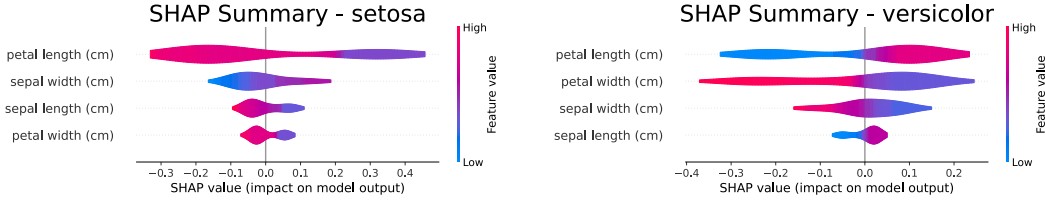

Figure 3: SHAP summary plots for Setosa and Versicolor predictions.

## 4.3 LEARNING SALIENCY MASK WITH QPERT

To further quantify the importance of each input feature, we calculate the QPERT saliency mask, shown in Figure 5. This visualization directly reflects the perturbation sensitivity of each feature in the quantum model. This also complements the class summaries by showing that while individual classes can prioritize certain characteristics (e.g., petal length for *Setosa* as shown in Figure 4 ), the model as a whole treats all characteristics as comparable when evaluated across the entire test set.

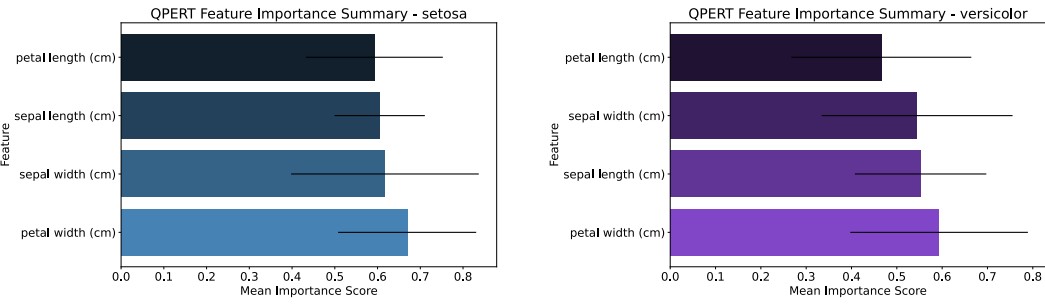

Figure 4: QPERT summary plots for Setosa, and Versicolor predictions.

To better understand how the quantum classifier distinguishes between classes, we analyzed the feature importance scores generated by QPERT for each class in the Iris dataset. For *Setosa*, the model placed greatest emphasis on petal width, with sepal features contributing to a lesser extent, reflecting the clear separability of this class. In contrast, *Versicolor* exhibited a more balanced dependence on both the petal and sepal characteristics consistent with its greater overlap with *Virginica* (Figure 11). These differences in feature importance profiles highlight the model's ability to adapt its decision-making strategy based on class-specific patterns, using different combinations of features to achieve an accurate classification. These results illustrate how QPERT characterizes the local sensitivity of the quantum model's output to changes in the classical input features. Although this section does not directly analyze fidelity, entanglement, superposition, or circuit-internal quantum states, the perturbation-based approach nonetheless provides an interpretation consistent with the decision boundaries induced by the quantum circuit.

## 4.4 LOSS TRENDS

Figure 6 shows the evolution of the loss functions during progressive training, plotted at an interval of 250 iterations. This staged approach prevents the optimizer from being overwhelmed by competing objectives early in the training and allows the model to gradually refine explanations

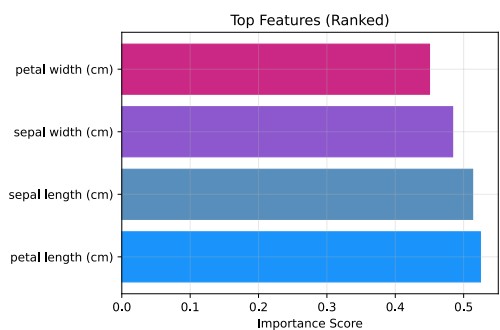

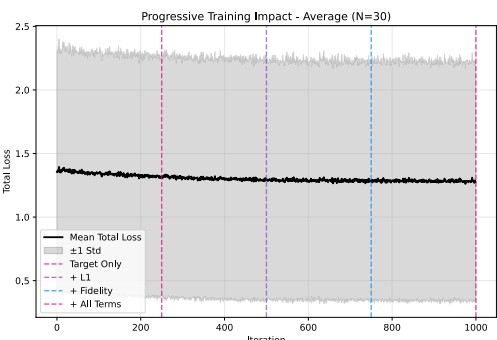

Figure 5: Global feature importance (QPERT saliency) aggregated across the entire test set. The bars show mean saliency per feature. This global summary complements the per class plots (Figure 4); while individual classes can emphasize specific features, the model's overall behavior reflects the ranked importance shown here.

Figure 6: The progressive training impact plot displays the total loss for all 30 test instances at each step of 250 iterations. While the improvement is marginal, it still influences other objectives such as fidelity or structural constraints, and highlights the challenge of achieving stable optimization in this setting.

with increasing complexity. The target loss is introduced from the 0th iteration. It represents the negative log-likelihood loss and encourages the model to assign high probabilities to correct classes (targets). This is followed by the introduction of L1 regularization loss, at the 250th iteration, which steadily declines, demonstrating sparsity in the perturbation mask. At 500th iteration, fidelity loss is introduced which remains close to zero across iterations, indicating that the perturbations introduced by QPERT preserve the original quantum state representations. Thereafter, both entanglement loss and superposition loss are introduced at 750th iteration. Entanglement loss converges rapidly, suggesting that QPERT maintains inter-feature quantum correlations essential to the QCL circuit's expressiveness. The superposition loss remains relatively stable, implying that the model avoids collapsing into overconfident deterministic states and preserves quantum uncertainty. In Figure 6, the plots help explain how each phase influences overall model convergence. Finally, these trends confirm that QPERT not only enhances interpretability but does so while adhering to key quantum mechanical principles.

Figure 7 presents a six-line graph that tracks the evolution of different loss metrics over 1000 training iterations for QPERT. The top left graph shows the total loss across five instances, each following a distinct trajectory, indicating variability in convergence behavior. The top center graph illustrates the average target loss with its standard deviation, showing a steady decline, suggesting effective optimization toward the target. The top right graph displays the L1 loss, which remains low and stable, implying minimal deviation in the learned representations. The bottom row focuses on interpretability-related losses: fidelity loss (bottom left) decreases gradually, indicating improved alignment with model behavior; entanglement loss (bottom middle) drops early and stabilizes, suggesting successful disentanglement; and superposition loss (bottom right) also trends downward, reflecting reduced overlap in explanatory components. Together, these curves offer a comprehensive view of how different aspects of QPERT's performance evolve during training.

## 4.5 HYPERPARAMETER STUDY

To optimize QPERT's performance and ensure robust explanation quality, we conduct a systematic hyperparameter study using grid search across all loss coefficient combinations. The hyperparameter exploration targets five key coefficients: target coefficient, L1 regularization coefficient, superposition coefficient ($L_{superposition}$), fidelity coefficient ($L_{fidelity}$), and entanglement coefficient ($L_{entanglement}$). Our grid search methodology generates all possible combinations within predefined ranges for each coefficient: target coefficient $\in \{0.5, 1.0, 2.5\}$, L1 regularization coefficient $\in \{0.05, 0.10, 0.30\}$, superposition coefficient $\in \{0.10, 0.20, 0.50\}$, fidelity coefficient $\in \{0.25, 0.5, 1.0\}$, and entanglement coefficient $\in \{0.1, 0.3, 0.6\}$. To maintain computational fea-

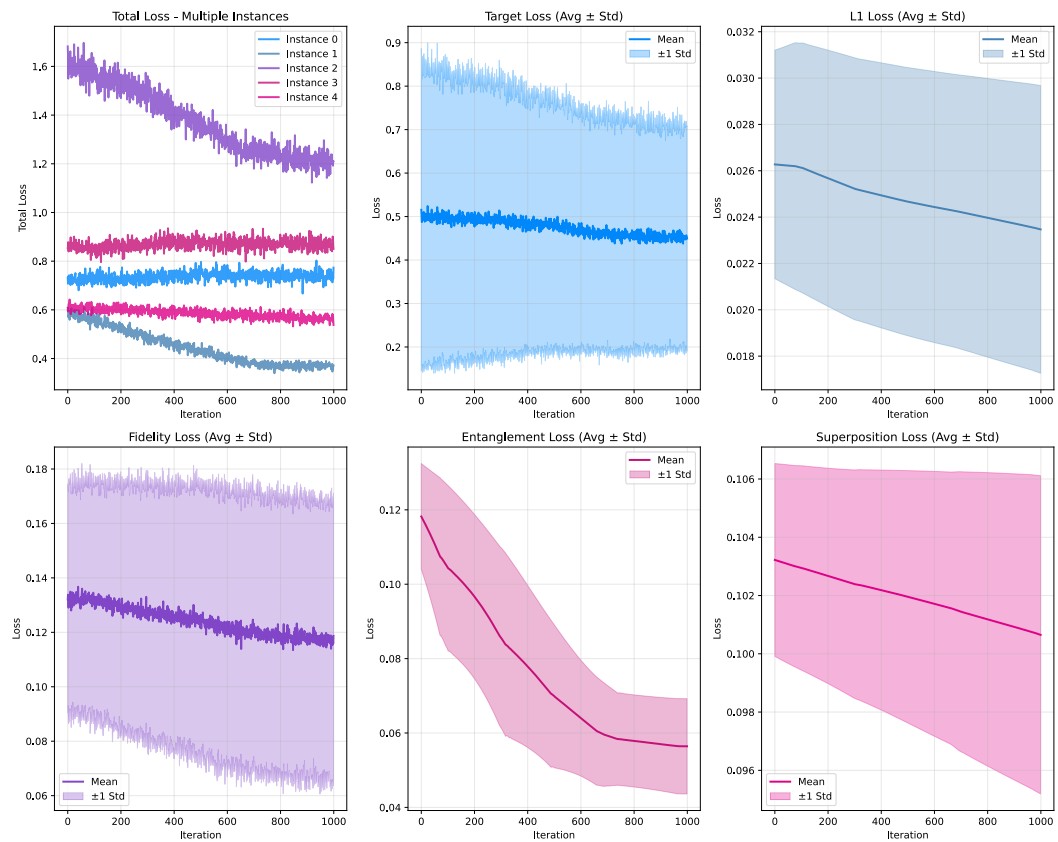

Figure 7: QPERT Loss curves illustrating the evolution of multiple loss components during training for five randomly selected instances over 1000 iterations.

sibility, we limit the search to a maximum of 20 trials, applying random sampling when the total number of combinations exceeds this threshold. For each configuration, we train the model using the quantum saliency generation function and evaluate performance based on the total loss, which serves as our primary optimization criterion. The search process systematically explores the hyperparameter space, tracking the best-performing configuration that achieves the lowest total loss while maintaining model stability. This comprehensive approach ensures that QPERT operates at optimal performance levels, with each loss component properly balanced to achieve high-quality saliency mask. The grid search results provide valuable insights into the sensitivity of different loss terms and guide the selection of coefficients that maximize explanation fidelity while maintaining computational efficiency across diverse time-series datasets.

The optimal configuration identified through this process yields the following coefficient values: target coefficient = 1.0, L1 regularization coefficient = 0.05, superposition coefficient = 0.1, fidelity coefficient = 0.3, and entanglement coefficient = 0.2. These values reflect the relative importance of each loss component in achieving high-quality explanations, with notably higher weights assigned to fidelity and entanglement terms, underscoring their critical role in maintaining explanation faithfulness and structural coherence.

## 4.6 Area Under the Curve(AUC)

**AUC-Difference**: A unified metric for evaluating saliency mask in QCL Models. Evaluating the quality of saliency mask in QCL models remains a critical challenge, particularly in the absence of ground-truth annotations for spatial-temporal relevance. To address this, we leverage perturbation-based evaluation a principled approach where the model's predictive confidence is monitored under

systematic deletion or insertion of input data ranked by their estimated importance Petsiuk et al. (2018).

**AUC-Deletion**: Deletion-based evaluation involves progressively removing the most important input data as identified by the saliency mask. The underlying hypothesis is that an informative saliency mask will assign higher scores to data segments that are crucial for the model's prediction. As these are deleted, the model's confidence in its original prediction should decline sharply. This behavior is captured by computing the Area Under the Deletion Curve (AUDC), which quantifies the degradation in prediction confidence as input data segments are removed in descending order of importance. A lower AUDC indicates a more faithful explanation, as it implies greater sensitivity of the model to the removed data segments.

**AUC-Insertion**: Insertion-based evaluation, in contrast, starts from a baseline input typically constructed as the mean of the opposite class and sequentially restores the most important data segments from the original instance. The Area Under the Insertion Curve (AUIC) quantifies the rate at which the model regains confidence in its original prediction as these data segments are reintroduced. A high AUIC suggests that the identified data segments effectively reconstruct the evidence needed for the prediction, again reflecting a high-quality saliency mask.

To consolidate these complementary perspectives, we propose the AUC-Difference metric, defined as:

$$\text{AUC-Difference} = \text{AUIC} - \text{AUDC} \tag{12}$$

This unified metric integrates both deletion and insertion dynamics, capturing the extent to which the model both depends on (low AUDC) and can be reconstructed by (high AUIC) the identified salient data segments. The ideal saliency mask would achieve an AUC-Difference close to 1.0, corresponding to AUIC nearly 1.0 and AUDC nearly 0.0. This formulation directly aligns with the model's internal decision boundaries, enabling a model-centric evaluation that minimizes the reliance on subjective human annotation or task-specific heuristics.

**Implementation Details** For deletion, we replace each data segment to be deleted with the corresponding data segments from the mean feature vector of an opposite class. This strategy ensures a semantically plausible input that remains within the data manifold, while systematically eliminating evidence relevant to the predicted class. For insertion, we begin with the opposing class mean and iteratively substitute the original data segments ranked by importance back into the baseline. This gradual reintroduction allows for fine-grained assessment of how evidence accumulation influences prediction confidence. By incorporating both insertion and deletion in a single, interpretable measure, AUC-Difference offers a robust, model-aligned, and annotation-free framework for evaluating explanation fidelity in QCL settings.

## 4.7 ABLATION STUDY

To empirically validate the effectiveness of different interpretability techniques for Quantum Circuit Learning models, we performed an ablation study using the AUC-Difference metric. We compared four explanation methods: **SHAP**, **QPERT** (our proposed method), **LIME**, and a **Random Baseline**. Each method was evaluated based on its ability to identify salient input features that align with the model's internal decision logic, as quantified by their Insertion AUC, Deletion AUC, and the resulting AUC-Difference.

Table 1: Ablation Study: Saliency Quality Across Explainers

| Explainer | AUIC | AUDC | AUC-Diff | Rank |
|---|---|---|---|---|
| SHAP | 0.529 | 0.455 | **0.074** | 1 |
| **QPERT** | 0.496 | 0.461 | **0.035** | 2 |
| LIME | 0.479 | 0.513 | -0.034 | 3 |
| Random Baseline | 0.449 | 0.502 | -0.053 | 4 |

The results in Figure 8 highlight key insights into the effectiveness of each method. These plots visualize how model confidence evolves as important input segments are removed or reintroduced,

offering intuitive support for the quantitative AUC metrics. SHAP emerged as the top-performing method, achieving the highest AUC-Difference and confirming its strong alignment with the QCL model's decision boundaries. QPERT method demonstrated competitive performance, outperforming both LIME and the Random Baseline. Notably, LIME yielded a negative AUC-Difference, suggesting that its explanations may not reliably reflect the model's behavior. The Random Baseline, as expected, produced a near-zero AUC-Difference, validating its role as a control.

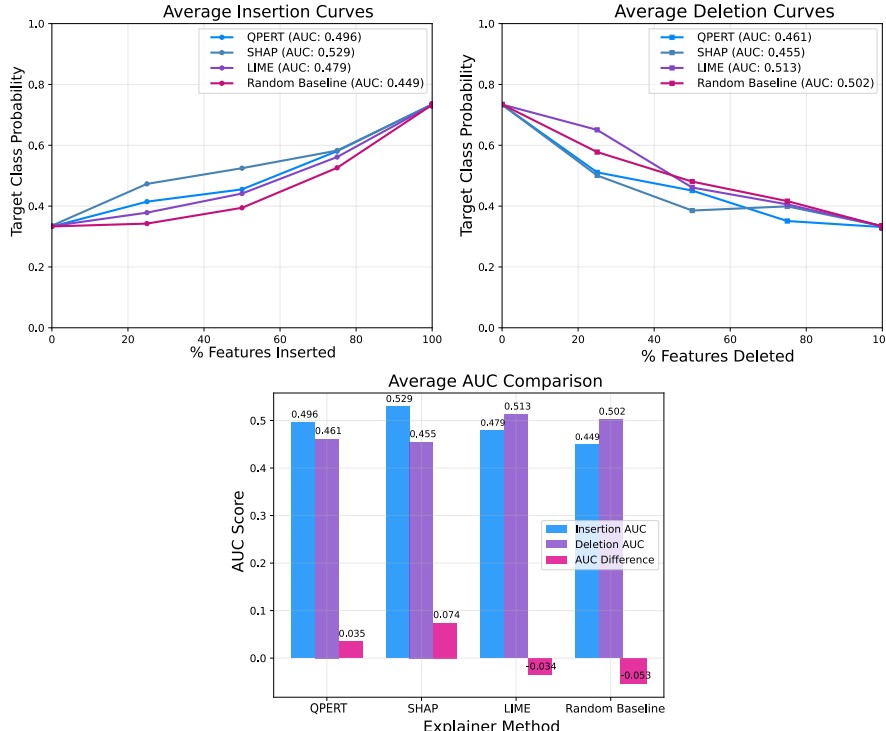

Figure 8: Insertion and Deletion Curves for SHAP, LIME, QPERT and a Random baseline.

## 5 CONCLUSION

In this work, we introduced QPERT, the first perturbation-based interpretability framework specifically designed for QCL models. QPERT is grounded in quantum-aware design principles, ensuring that explanation generation respects and preserves fundamental quantum properties such as fidelity, entanglement, and superposition, thereby maintaining the integrity of quantum information throughout the interpretability process. Local analysis using LIME confirmed the dominance of petal-related features, particularly petal length, in Virginica classification, aligning with domain knowledge. SHAP-based global analysis validated the QCL model's attribution patterns across multiple classes. QPERT's saliency masks provided quantitative insights into feature importance, emphasizing relevant inputs while discounting noise. Moreover, convergence patterns of our quantum-specific loss functions confirmed that QPERT achieves interpretability without degrading quantum model behavior. Throughout training, fidelity loss remained low, entanglement correlations were preserved, and superposition states were maintained highlighting the method's compatibility with quantum constraints. Our empirical evaluation through ablation study confirms that QPERT produces faithful, semantically coherent explanations.

Overall, QPERT provides a principled, interpretable, and consistent approach to explainable QML.

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

## A  DATA PREPARATION

We use the Iris dataset from the `scikit-learn` library, which contains 150 samples, each described by 4 numerical features and assigned to one of 3 classes. This dataset is widely adopted in quantum machine learning research due to its small size and multi-class nature, making it compatible with current quantum simulation constraints. To prepare the data for input into quantum circuits, we normalize all features to the range $[0, 1]$ using the `MinMaxScaler`, ensuring consistent scaling for quantum gate parameterization. Class labels are transformed into one-hot vectors using the `LabelBinarizer`, which is compatible with the softmax output and cross-entropy loss function used during training. The dataset is partitioned using stratified sampling with a fixed random seed to preserve class balance and ensure reproducibility. We allocate 64% of the data to training, 16% to validation, and the remaining 20% to testing. Throughout preprocessing, we retain feature and class names to support interpretability analyses and visualizations in subsequent components of the pipeline.

## B    Training Configuration and Validation

The quantum classifier is implemented as a variational quantum circuit (VQC) using the Qiskit framework. The circuit operates on 4 qubits and has a depth of 3, with entangling layers inserted between parameterized single-qubit rotations. The model currently encodes classical Iris input data into quantum states using Angle encoding, but this can be generalized by replacing it with a modular function, allowing compatibility with other encoding schemes such as Amplitude encoding. Each forward pass is executed using 2000 measurement shots to ensure statistical stability in the estimated output probabilities. Optimization is performed using mini-batch gradient descent, where gradients are computed via the parameter-shift rule. The training loop uses a batch size of 4 and runs for up to 300 epochs. To prevent overfitting and ensure convergence, we employ early stopping with a patience of 20 epochs. If the validation loss does not improve for 12 consecutive epochs, the learning rate is decayed exponentially by a factor of 0.97. Training is terminated early if the validation loss falls below a threshold of 0.1. The initial learning rate is set to 0.3. Both training and validation losses are monitored throughout the process, and the best-performing model parameters, as measured by validation loss, are retained for final evaluation and explanation.

## C    Simulation Environment

We develop a hybrid simulation framework that integrates quantum circuit execution with classical data preprocessing and optimization. Quantum components are implemented using Qiskit's `QuantumCircuit` class and executed on the `AerSimulator` backend, which provides a noise-free simulation of quantum circuits. Input features are encoded into quantum states using rotation gates, such as $R_y$ and $R_z$, with each qubit corresponding to one feature. The variational circuit comprises several layers of parameterized single-qubit rotations interleaved with entangling gates (e.g., CNOT). Measurements are performed in the computational basis, and the resulting bitstrings are used to estimate class probabilities. The predicted class is determined by aggregating measurement outcomes and applying a modulo-3 mapping scheme, which evenly distributes measurement states across the three output classes.

Classical components are implemented using NumPy, PyTorch, and `scikit-learn`, and are responsible for data preprocessing, batching, loss computation, and evaluation. The training loop incorporates the parameter-shift rule to compute gradients analytically, and uses the cross-entropy loss to compare predicted probabilities with one-hot encoded labels. Evaluation includes computing classification accuracy, loss on the test set, and the confusion matrix to assess class-wise prediction behavior.

## D    Quantum Explanation Framework

To address the interpretability challenges associated with quantum machine learning models, we develop a local explanation framework based on instance-wise feature perturbation. Given a trained quantum classifier and a test input, our method, QPERT, constructs a feature importance mask by solving a constrained optimization problem that identifies the minimal subset of input features responsible for the model's prediction.

The objective function balances multiple loss terms. The target loss is the negative log-likelihood of the original class under perturbed input, promoting faithfulness to the original decision. Sparsity is enforced through an $L_1$ penalty on the mask vector. To ensure that the model output remains consistent after perturbation, we include a fidelity loss term that minimizes the divergence between original and perturbed output distributions. This divergence is measured using either Jensen-Shannon divergence or $L_2$ norm, depending on the stability of gradients. We also introduce an entanglement loss that penalizes masks which result in high mutual information between distant qubits, encouraging disentangled representations. Finally, a superposition loss term promotes discrete mask configurations by minimizing the participation ratio and entropy of the mask distribution, encouraging sharper and more interpretable importance scores.

Optimization is performed using the Adam optimizer, and the training of the mask follows a phased schedule in which auxiliary losses other than target loss are gradually introduced to ensure stable

702
703
704
705
706
707
708
709
710
711
712
713
714
715
716
717
718
719
720
721
722
723
724
725
726
727
728
729
730
731
732
733
734
735
736
737
738
739
740
741
742
743
744
745
746
747
748
749
750
751
752
753
754
755

convergence. The explanation method is applied independently to each test instance, resulting in a sparse feature importance vector that highlights the features most influential to the prediction.

## E    EVALUATION METHODOLOGY

Model performance is evaluated using standard metrics, including classification accuracy, average cross-entropy loss on the test set, and class-wise confusion matrices. For interpretability evaluation, we compare QPERT explanations to those produced by SHAP and LIME, two widely used post-hoc explanation methods. To assess the fidelity and relevance of explanations, we conduct ablation studies in which features are progressively inserted or removed based on their ranked importance scores. The change in model confidence or predicted probability is tracked, and the AUC of the confidence change is computed. A larger AUC difference indicates that the explanation method correctly identifies features that meaningfully impact the model's prediction.

In addition to quantitative metrics, we assess explanation sparsity, convergence behavior, and stability across multiple runs. We also examine whether QPERT explanations yield better alignment with model decision boundaries compared to classical methods, particularly in the presence of quantum-specific interactions such as entanglement. All evaluations are performed using the retained model checkpoint selected based on validation performance.

To address the complexity of our QCL model, on a quantum computer, loss evaluation for a 4 qubit QML model scales as $(2p + 1) \times$ shots, where $p = 12L + 8$ for circuit depth L. For L=3 and 2,000 shots, this is about 178,000 circuit executions per step, plus hardware latency and noise. On a classical computer (the current scenario), loss and gradients are computed in a single pass with polynomial complexity, making it orders of magnitude faster.

## F    SUPPLEMENTARY EXPLANATION PLOTS

To complement the global interpretability analysis in Section Results, we provide class-specific explanation plots for the Virginica class that were omitted from the main text due to space constraints.

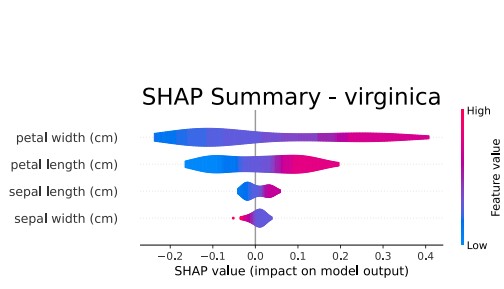

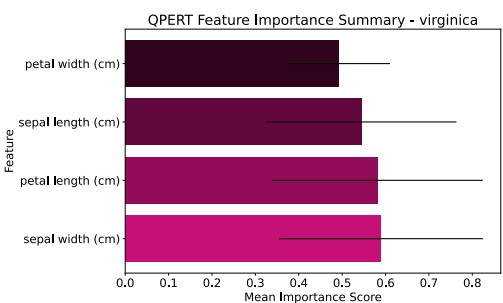

Figure 9: (a) SHAP summary plot for Virginica predictions. This plot highlights global feature importance and the direction of impact on the model's output using SHAP values.

Figure 10: (b) QPERT summary plot for Virginica predictions. This plot shows the estimated global feature importance using the quantum-inspired QPERT method.

Figure 11: Supplementary explanation plots for the *Virginica* class using SHAP (a) and QPERT (b) methods.

## G    FUTURE WORK

Although current QPERT evaluations focus on the IRIS dataset, future work will explore its applicability to more complex and standardized datasets such as MNIST, time-series data, medical imaging, and financial portfolio assessment. These domains present diverse challenges and will help validate QPERT's generalizability across different data modalities. We also plan to make the underlying QCL

model compatible with GPU acceleration and leverage Aer GPU simulators to improve scalability and performance on larger quantum circuits with more qubits. Preliminary experiments have shown consistent interpretability results, but further validation is needed under realistic quantum noise conditions. To address noise and hardware limitations, future iterations of QPERT may explore error mitigation techniques and potentially noise-aware perturbation strategies to ensure reliable explanations even in imperfect quantum environments. Establishing standardized benchmarks and metrics for quantum interpretability will further facilitate fair comparisons with emerging methods and promote reproducibility in the field.

