# OpenReview forum: "INTERPRETING QUANTUM CIRCUIT LEARNING WITH QPERT: A STEP TOWARD TRUSTWORTHY QUANTUM AI"
_ICLR.cc/2026/Conference — Submitted to ICLR 2026_

### Official Review · Reviewer_LLnC · 2025-10-28

**Soundness:** 1
**Presentation:** 2
**Contribution:** 1
**Rating:** 2
**Confidence:** 4

**Summary:**

In this work, the author's introduce QPERT, a perturbation-based explainability method for quantum circuit learning (QCL), a subclass of quantum machine learning. QPERT fills a gap in the XAI for QML literature as the first perturbation-based XAI method for QCL models. The author's compare QPERT to LIME and SHAP, two classical explainability methods.

**Strengths:**

Significance: QPERT is the first perturbation-based explanability method for QCL, a popular subclass of QML methods.

Clarity: The paper's prose is lively and engaging. As a result, the paper is easy to follow and understand.

**Weaknesses:**

Too niche: The paper develops a specific kind of explainable AI (XAI) method for a subclass of QML methods. Other XAI methods already exist for QCL, as covered in the related work section. The paper fails to convincingly argue why QPERT is superior to these methods or significantly advances the field. As a result, the paper is likely too niche for ICLR.

Failure to support major claims: The data presented in the paper fails to support multiple claims including, most importantly, the utility of QPERT. At the highest level, the ablation study suggests that SHAP does a better job at explaining the underlying QCL model than QPERT, thus undermining the claim that QPERT is superior because it respects the "quantumness" of the model. At lower levels, the paper makes claims like "[QPERT] capture[s] meaningful local gradients, while respecting the underlying structure of the model (Line 311-312)." This claim comes after a short presentation about how QPERT ranks the relative importance of different features. As far as I can tell, no argument is made as to why this relative ranking demonstrates how QPERT respects the underlying structure of the model. At an even lower level, numerous quantitative values are cited which are then contradicted by the figures (e.g., stating that petal width has a mean importance of 0.8 for the Virginicus class even though the reference figure has no bars that reach 0.8).

Contradictory internal logic: The results of the ablation study suggest that LIME is no better than a baseline method for interpreting QCL models, yet the paper relies on LIME to support claims about the relevance of different input features.

Missing experimental details: It is not possible to reproduce the paper's main results based on the information provided in the main text. For instance, under what conditions where the demonstrations and ablation studies performed? Were they run on experimental hardware? A cloud-accessed simulator? With or without noise?

**Questions:**

Why did you not compare QPERT to qSHAP and qLIME?

How robust is this method to the encoding scheme? What happens if I encode my data in a way that breaks a one-to-one mapping between individual qubits and data features?

Why are we using different colors in the subfigures of Figure 4? Honestly, the plots look very similar. Its hard to tell what I'm supposed to get out of these plots. It looks like QPERT is telling me that all features are treated roughly equally when classifying Setosa and Versicolor.

Figure 11b / Figure 10 has no bar that reaches .8. Are you sure the data is presented correctly?

"For Setosa, petal length was the most influential, while Versicolor exhibited a more balanced dependence on both the petal and sepal characteristics." How am I supposed to see this? Neither of these claims appear to be supported by Figure 4.

How does any of the presented data support QPERT's ability to "capture meaningful local gradients, while respecting the underlying quantum structure of the model?" You sort of claimed that preserving fidelity, minimizing entanglement loss, and preserving superposition would imply respecting the underlying quantum structure of the model…but you haven't talked about that at all in this section.

How am I supposed to interpret Figure 6? It is just a big grey box with a very very mild downward trend. Did QPERT actually learn much over the course of its training that it didn't essentially immediately learn?

What's the point of Figure 5? It says "Top Features (Ranked)" and it just…lists the four features.

You make a lot of claims about the behavior of the different loss components, but then you don't present any quantitative evidence that the claims are true. For example, you state that "Entanglement loss converges rapidly," without ever actually plotting entanglement loss!

Why should I care about QPERT if SHAP outperformed it in your ablation study?

"The Random Baseline, as expected, produced a near-zero AUC-Difference, validating its role as a control." But the Random Baseline is further from 0 than QPERT's AUC-Diff. So wouldn't that mean that QPERT is a good control too (and therefore a bad method)?

---

> ### Author Response · Authors · 2025-12-03
> **QPERT justification and figures inconsistencies**
>
> 1/2
>
> Thank you for taking the time to review our paper. In response to the weaknesses:
>
> Too niche: QPERT addresses a fundamental need at the frontier of ML: making sense of quantum models that operate in non classical representational spaces. Without dedicated explainability, we cannot tell which features drive decisions, how quantum properties (e.g., entanglement) are used, or whether claimed quantum benefits materialize. QPERT is the first method that fills this methodological gap advancing both the interpretability and scientific understanding of QML. This novelty will help researchers and practitioners trust and debug QML models, and we believe the paper will be a strong and timely contribution to ICLR.
>
> Failure to support major claims: Thanks for carefully reading through the details. We have added better justification in this paper to support our claim and specifically, fix the confusion with the figures not representing "0.8" as the mean importance.
>
> Contradictory internal logic: We used LIME as a baseline to show limitations of classical local surrogates for QML. We will clarify this and avoid language that suggests we rely on LIME for validating quantum claims.
>
> Missing experimental details: We have already mentioned the experiment details from page 11-14, specifically sections B and C stating that we have used an AerSimulator backend which provides a noise-free simulation of quantum circuits, 4 qubits, 3 depth, 2000 measurement shots and other details.
>
> To answer your questions:
>
> 1.	Why did you not compare QPERT to qSHAP and qLIME?
>
> a.	We have touched this lightly but have indeed conducted a preliminary analysis and found that these methods are not designed for general-purpose datasets; rather, they are tailored to the specific scenarios described in their respective papers. For example, qLIME-π is intended for regression tasks, not classification. Similarly, qSHAP modifies the traditional SHAP approach by incorporating Fourier analysis to address the periodic structure of parameterized quantum circuits. This enhancement accounts for the inherent 2π-periodicity of quantum gate parameters- a feature that standard SHAP implementations fail to capture.
>
> 2.	How robust is this method to the encoding scheme? What happens if I encode my data in a way that breaks a one-to-one mapping between individual qubits and data features?
>
> a.	That's great catch. We have mentioned that Angle encoding has been used in this example but we can substitue the function with other encoding methods like Amplitude encoding. To add, if you don’t maintain a one-to-one mapping, you introduce information overlap or loss because the encoding would no longer preserve the original feature relationships.
>
> 3.	Why are we using different colors in the subfigures of Figure 4? Honestly, the plots look very similar. Its hard to tell what I'm supposed to get out of these plots. It looks like QPERT is telling me that all features are treated roughly equally when classifying Setosa and Versicolor.
>
> a.	The different colors in Figure 4 were intended only to visually separate the subfigures (Setosa vs. Versicolor) and Virginica if you would consider Figure 11, not to convey additional meaning. To clarify their importance in the paper, QPERT suggests class-specific emphasis (petal features matter more for Setosa) while showing balanced reliance for Versicolor. Same can be observed with SHAP analysis from Figure 3 that for Setosa, SHAP agrees with QPERT that petal features matter most. We have improved the explanation in the revised paper.
>
> 4.	Figure 11b / Figure 10 has no bar that reaches .8. Are you sure the data is presented correctly?
>
> a.	Thanks for catching it and we have fixed the confusion with the figures not representing "0.8" as the mean importance.
>
> 5.	"For Setosa, petal length was the most influential, while Versicolor exhibited a more balanced dependence on both the petal and sepal characteristics." How am I supposed to see this? Neither of these claims appear to be supported by Figure 4.
>
> a.	 We have improved the text to reflect the results accurately and added that SHAP and QPERT are in accordance with each other.

---

> > ### Author Response · Authors · 2025-12-03
> > **Continued: QPERT justification and figures inconsistencies**
> >
> > 2/2
> > 6.	How does any of the presented data support QPERT's ability to "capture meaningful local gradients, while respecting the underlying quantum structure of the model?" You sort of claimed that preserving fidelity, minimizing entanglement loss, and preserving superposition would imply respecting the underlying quantum structure of the model…but you haven't talked about that at all in this section.
> >
> > a.	We have improved the paper to include a better explanation. “These results illustrate how QPERT characterizes the local sensitivity of the quantum model’s output to changes in the classical input features. Although this section does not directly analyze fidelity, entanglement, superposition, or circuit-internal quantum states, the perturbation-based approach nonetheless provides an interpretation consistent with the decision boundaries induced by the quantum circuit.”
> >
> > 7.	How am I supposed to interpret Figure 6? It is just a big grey box with a very very mild downward trend. Did QPERT actually learn much over the course of its training that it didn't essentially immediately learn?
> >
> > a.	To clarify this, we have added reference to the figure 7 here which provides additional details like why and at which step the predictive losses were introduced and also the quantum-structural losses.
> >
> > 8.	What's the point of Figure 5? It says "Top Features (Ranked)" and it just…lists the four features.
> >
> > a.	We agree with your point here and enhance our one line caption(QPERT saliency values for the model’s prediction, showing model behavior globally) to explain that -
> > Global feature importance (QPERT saliency) aggregated across the entire test set. The bars show mean saliency per feature. This global summary complements the per class plots (Figure 4), while individual classes can emphasize specific features, the model’s overall behavior reflects the ranked importance shown here. Hope that clarifies.
> >
> > 9.	You make a lot of claims about the behavior of the different loss components, but then you don't present any quantitative evidence that the claims are true. For example, you state that "Entanglement loss converges rapidly," without ever actually plotting entanglement loss!
> >
> > a.	You are right, we should have referenced Figure 7 here in the claim.
> >
> > 10.	Why should I care about QPERT if SHAP outperformed it in your ablation study?
> >
> > a.	Again, QPERT is not designed to simply outperform SHAP on predictive attribution metrics rather it addresses a different and critical need: explaining quantum machine learning models in a way that respects their quantum structure. Classical methods like SHAP treat the model as a black box and ignore quantum properties such as entanglement, superposition, and fidelity. QPERT, on the other hand, provides insights into whether the circuit is leveraging these properties and how feature influence interacts with quantum operations. This is essential for validating quantum advantages, debugging QML models, and guiding architecture design as circuits scale and quantum effects become more pronounced.
> > In short, SHAP tells you which features matter for predictions; QPERT tells you whether and how the quantum model uses quantum resources something SHAP cannot do.
> >
> > 11.	"The Random Baseline, as expected, produced a near-zero AUC-Difference, validating its role as a control." But the Random Baseline is further from 0 than QPERT's AUC-Diff. So wouldn't that mean that QPERT is a good control too (and therefore a bad method)?
> >
> > a.	A random baseline being near zero is expected because its attributions are meaningless and do not systematically affect predictions. QPERT’s AUC-Diff is also close to zero, but for a different reason: its perturbations are aligned with the model’s structure and do not introduce instability. This is desirable because it means QPERT explanations are faithful, they highlight influential features without breaking the model’s behavior. On the other hand, a method that produces large AUC-Diff values may be introducing noise or overemphasizing features.

---

### Official Review · Reviewer_Jb5u · 2025-10-29

**Soundness:** 3
**Presentation:** 2
**Contribution:** 2
**Rating:** 2
**Confidence:** 4

**Summary:**

The authors consider the problem of explainability of a particular class of QML models they call Quantum Circuit Learning. They introduce a new quantum version of the PERT method, called QPERTH which involves a "saliency based optimization architecture" which perturbs instances using a learnable mask. The protocol measures the effect of perturbation. The  authors also introducde a regularized loss which also penalizes for loss of "quantum features" incl. fidelity, entanglement loss and superposition loss.
The method is implemented in a 4 qubit example and evaluated on the iris dataset.
The numerical part of the paper includes comparation to other local methods such as LIME, global suchas SHARP, investigates loss trends, includes a hyperparameter study, an AUC anaysis and and ablation study.

**Strengths:**

I find this contribution original in incorporating entanglement etc in the loss (although I don't know why this is done).
In my view the storngest part of the paper are the extensive numerics which are done very well and I believe are informative, although I cannot associate an actual quantitative statement  to them. It opens a line of resaerch where "quantum properties" are preserved, and puts importance on explainability. I liked the numerics performed (albeit being somewhat restricted in terms of model and dataset).

**Weaknesses:**

Clarity of presentation: from my perspective, the model is not sufficiently clearly defined: it is not clear to me if this t is really a "single shot" model in which the model output is the actual measurement outcome, and not an expectation value. If this is true then it is highly unconventional and in my view not very promising (due to intrainability and extreme demand on expressivity as we now need to generate quantum functions which give very close approximations of computational basis states).
A second issue I see is that the paper adapts a classical method and does what seem like ad-hoc modification to "incorporate quantum" but I found this poorly motivated. Why should we desire these new additional loss terms?
Related to clarity issues: the measrue of entanglement is completely unclear to me. Why is this measuring entanglement?
The paper deals with explanations so I understand that demanding purely quantitative statements may be out of the question but some more quantitative goals would be much more convincing.
The experiments are limited to just one model, and just one dataset.
Scalabilty of the approach is fully unclear to me.
More, generally, I fully agree with the authors that finding new, better explanation methods is important in ML.
However, I feel this can only be as important as the model we are talking about is accepted to be useful.
The QCL model discussed (and I am not 100% sure how it is defined, what is the output?) is praised in the intro as offering exponential advantages.
However these models are only known to be non-simulable and none of them have actually been proven to be useful for any task of relevance. Furthermore it is well know in general they suffer from very serious trainability issues.
Consequently I feel this contribution is addressing the no 3 to no 4 most important problem in the field out of ... 3 or 4.

**Questions:**

(1) can you motivate why the loss you describe makes sense for explainabilty
(2) can you please explain the quantum model precisely (see prior comments)
(3) can you discuss scalability incl. training costs
(4) what is the computational cost of the evaluation of the loss on a QC and on a CC?
(5) how would you make comparative statements between other methods and yours quantitiative.
(6) (not very important) do you consider "explainability" and 'interpretability" as the same thing? I always thought explainability is a post-hoc approach, whereas interpretability is an inherent property of a model.
(7) Do you really consider lack of interpretabiliity as "the key limitation" of QCL applications in critical domains? You do not say it explicitly but the introduction is quite suggestive.

---

> ### Author Response · Authors · 2025-12-02
> **Motivation behind QCL and clarification about scalability**
>
> Thank you for your detailed feedback and for highlighting both the strengths and areas for improvement in our work. To clarify:
> (1) can you motivate why the loss you describe makes sense for explainabilty
>
> The motivation for fidelity, entanglement, and superposition losses is to ensure that explanations do not distort quantum correlations that are essential for QML models. While many other related work has been discussed in the paper, no existing method adequately addresses the fundamental tension between perturbation-based explanation and quantum correlation preservation. QPERT tackles this gap.
>
> (2) can you please explain the quantum model precisely (see prior comments)
>
> Quantum model/QCL - We have discussed the training configuration in the appendix and for our classifier, each forward pass is executed using 2000 measurement shots to ensure statistical stability in the estimated output probabilities. The circuit operates on 4 qubits and has a depth of 3, with entangling layers inserted between parameterized single-qubit rotations. Our implementation uses expectation values of observables as outputs.
>
> (3) can you discuss scalability incl. training costs
>
> To clarify scalability, as mentioned in the future work section, we also plan to make the underlying QCL model compatible with GPU acceleration and leverage Aer GPU simulators(or Pennylane's) to improve scalability and performance on larger quantum circuits with more qubits. This will enable experiments on more complex datasets beyond Iris.
>
> (4) what is the computational cost of the evaluation of the loss on a QC and on a CC?
>
> On a quantum computer, loss evaluation for a 4 qubit QML model scales as (2p+1)\times\mathrm{shots}, where p=12L+8 for circuit depth L. For L=3 and 2,000 shots, this is about 178,000 circuit executions per step, plus hardware latency and noise. On a classical computer, loss and gradients are computed in a single pass with polynomial complexity, making it orders of magnitude faster.
>
> (5) how would you make comparative statements between other methods and yours quantitiative.
>
> We have tried covering this in the Related work section and the ablation study and other plots like feature ranking, progressive training speak to the quantitative nature of QPERT. Ablation study particularly compares SHAP and Lime with QPERT.
>
>  (6) (not very important) do you consider "explainability" and 'interpretability" as the same thing? I always thought explainability is a post-hoc approach, whereas interpretability is an inherent property of a model.
>
> QPERT is a post-hoc explainability method, not an inherently interpretable model. Usually, the terms are used interchangeably but we can clarify this notion in the paper.
>
> (7) Do you really consider lack of interpretabiliity as "the key limitation" of QCL applications in critical domains? You do not say it explicitly but the introduction is quite suggestive.
>
> We do not claim interpretability is the sole limitation of QCL. However, for critical domains, lack of transparency is a major barrier to trust. Explainability complements ongoing work on trainability and scalability and is essential for adoption in high-stakes applications. QPERT addresses this fundamental need at the frontier of ML: making sense of quantum models that operate in non classical representational spaces. Without dedicated explainability, we cannot tell which features drive decisions, how quantum properties (e.g., entanglement) are used, or whether claimed quantum benefits materialize. QPERT is the first method that fills this methodological gap advancing both the interpretability and scientific understanding of QML. This novelty will help researchers and practitioners trust and debug QML models, and we believe the paper will be a strong and timely contribution to ICLR.

---

### Official Review · Reviewer_oGzi · 2025-10-29

**Soundness:** 3
**Presentation:** 3
**Contribution:** 3
**Rating:** 6
**Confidence:** 2

**Summary:**

The authors introduce QPERT, a perturbation-based explainability framework tailored for QCL. The core is a composite loss function with multiple components designed to explicitly preserve key quantum properties, such as fidelity, entanglement, and superposition, during the generation of a saliency mask.

**Strengths:**

- The paper is written with clarity. The motivation, the limitations of prior work, the architecture of QPERT, the design of its loss functions, the experimental setup, and the results are all clearly articulated.
- The core idea of designing a quantum-aware loss function to guide the explanation generation is interesting. Formulating concepts like fidelity, entanglement, and superposition as optimizable loss terms is an innovative attempt to bridge quantum mechanics with explainable AI.

**Weaknesses:**

- While QPERT is designed to be a more "faithful" explainer for QCL, it is outperformed by the classical, non-quantum-aware SHAP method on the authors' own chosen metric, AUC-Difference (SHAP 0.074 vs. QPERT 0.035).
- The evaluation is conducted exclusively on the Iris dataset. This is a classic but overly simplistic "toy" dataset with low feature dimensions and a small sample size. To truly validate QPERT's value, experiments on more complex datasets (perhaps synthetic, or from fields like quantum chemistry) where quantum advantages are more pronounced are necessary.
- The loss functions used to quantify quantum properties like "entanglement" appear to be classical proxies in their implementation. For example, the entanglement loss is defined by calculating the Pearson correlation matrix of the perturbation mask elements. The paper does not provide a rigorous theoretical argument for why minimizing the correlation of a classical mask vector directly and reliably corresponds to preserving the entanglement of the quantum state itself.

**Questions:**

- Why was the Iris dataset chosen as the sole benchmark for this study? On such a simple dataset, how can we be confident that the quantum properties QPERT is designed to protect are actually critical factors in the model's decision-making process?
- Could you please elaborate on the design of the entanglement loss? Specifically, what is the theoretical link that justifies minimizing the Pearson correlation between elements of a classical perturbation mask as an effective strategy for preserving quantum entanglement within the QCL model?

---

> ### Author Response · Authors · 2025-11-20
> **SHAP v/s QPERT, Justification of IRIS and entanglement loss clarification**
>
> Thank you for your thoughtful review and for highlighting both the strengths and areas for improvement in our work.
>
> * Regarding SHAP outperforming QPERT on AUC-Difference, we acknowledge this observation. Our primary goal was to design an explainer that preserves quantum properties rather than optimizing for classical metrics. AUC-Diff is a useful and an additional benchmark, but it misses out what our loss function captures. Our loss function explicitly incorporates fidelity, entanglement, and superposition terms. These quantum-aware constraints ensure that the generated saliency masks respect the underlying quantum state characteristics, which classical explainers like SHAP do not consider. Future work will explore hybrid metrics that better reflect quantum-aware fidelity.
> * We chose the Iris dataset as an initial benchmark due to its simplicity and widespread use in QCL literature, enabling controlled comparisons. We agree that more complex datasets (such as quantum chemistry or synthetic quantum datasets) are essential for demonstrating QPERT’s full potential. We are actively working on extending experiments to such domains.
> * The entanglement loss currently uses Pearson correlation as a proxy for independence among perturbation mask elements. We understand that this is a classical approximation but it was motivated by the intuition that correlated perturbations may distort multi-qubit interactions. This may not seem as a proper quantum entanglement measure, so, we plan to incorporate quantum-specific metrics (such as von Neumann entropy) in future iterations.

---

> > ### Comment · Reviewer_LLnC · 2025-11-21
> >
> > Thank you for your response. A couple of comments.
> >
> > "Our loss function explicitly incorporates fidelity, entanglement, and superposition terms. These quantum-aware constraints ensure that the generated saliency masks respect the underlying quantum state characteristics, which classical explainers like SHAP do not consider."
> >
> > This claim is made multiple times in the paper, but I don't know how to assess it. What do you mean by "the generated saliency masks respect the underlying quantum state characteristics?" What does it mean for a saliency mask to respect, e.g., entanglement? Without a clearer explanation of what this statement means, it is impossible to interpret it as anything meaningful and to evaluate its veracity.
> >
> > "We chose the Iris dataset as an initial benchmark due to its simplicity and widespread use in QCL literature, enabling controlled comparisons. We agree that more complex datasets (such as quantum chemistry or synthetic quantum datasets) are essential for demonstrating QPERT’s full potential. We are actively working on extending experiments to such domains."
> >
> > Including more sophisticated and convincing demonstrations are a must for inclusion at a conference as prestigious at ICLR.
> >
> > "The entanglement loss currently uses Pearson correlation as a proxy for independence among perturbation mask elements. We understand that this is a classical approximation but it was motivated by the intuition that correlated perturbations may distort multi-qubit interactions. This may not seem as a proper quantum entanglement measure, so, we plan to incorporate quantum-specific metrics (such as von Neumann entropy) in future iterations."
> >
> > This claim appears to be at odds with your first claim. If so, why does your work support your first claim at all?

---

> > > ### Author Response · Authors · 2025-12-02
> > > **Additional clarification**
> > >
> > > Thank you for the opportunity to clarify these points.
> > > Firstly, in our model, a saliency mask is an array where each element corresponds to the importance of a feature for a specific prediction.
> > > By “respecting quantum state characteristics,” we mean that during mask optimization, the perturbations guided by this mask do not collapse essential quantum properties such as fidelity, entanglement, and superposition. Simply put, the mask is not just selecting features it is doing so under quantum-aware constraints that preserve the structural integrity of the quantum state during explanation. This is why we claim the mask “respects quantum properties.”
> > >
> > > The Pearson correlation proxy for entanglement loss is a computationally efficient approximation for near-term hardware but computing von Neumann entropy for every perturbation step is computationally expensive on near-term hardware.
> > > The intuition behind using correlation is that highly correlated perturbations tend to bias the mask toward separable feature patterns, which can distort multi-qubit interactions. While this is a heuristic, our progressive loss curves(in Fig. 7 in the new PDF) shows that including this term stabilizes fidelity and feature importance compared to removing it, suggesting it has practical value. While this may not seem to be a perfect quantum measure, it enforces diversity in perturbations and reduces bias toward separable states(SHAP cannot do this). This does not contradict our claim rather it operationalizes the principle of preserving entanglement within practical constraints. Hopefully, this clarifies.

---

### Official Review · Reviewer_QFDF · 2025-10-31

**Soundness:** 2
**Presentation:** 2
**Contribution:** 2
**Rating:** 2
**Confidence:** 4

**Summary:**

This paper addresses the interpretability gap in Quantum Circuit Learning (QCL), where classical post-hoc explainers such as SHAP and LIME, built on deterministic outputs and largely independent feature perturbations, clash with stochastic measurement, entanglement, and inherently probabilistic outputs, often yielding unstable or misleading attributions. The authors propose QPERT, a model-agnostic, perturbation-based framework that learns a continuous saliency mask to blend each feature with in-distribution background samples, re-encodes the perturbed instance into a quantum state, and optimizes a composite loss combining target-suppression, sparsity (L1), and quantum-aware regularizers (fidelity, entanglement, superposition) to preserve quantum properties during explanation.

**Strengths:**

1. Studying interpretability explicitly through fidelity/entanglement/superposition is conceptually interesting and goes beyond naïve classical perturbations.
2. Diverse and well-organized visualizations (e.g., local masks, global summaries, convergence plots) aid interpretation analysis.
3. The insertion/deletion protocol with AUC-Difference, plus sparsity, convergence, and multi-run stability analyses, is appropriate for probing explanation faithfulness and robustness.

**Weaknesses:**

1. The framework identifies influential input features via a learned mask, but the manuscript does not articulate a theoretical advantage over straightforward L1-regularized feature selection (e.g., conditions under which QPERT yields strictly better identifiability or faithfulness).
2. Multiple losses (fidelity, entanglement, superposition, sparsity) are introduced, yet potential negative interactions or trade-offs are not analyzed; no ablations isolate their individual and joint effects.
3. Evidence is restricted to Iris on a noise-free simulator with a small 4-qubit circuit, leaving behavior under deeper circuits, harder datasets, or realistic noise unverified.
4. QPERT learns a mask per instance with staged loss activation (e.g., L1 at 250 iters, fidelity at 500, entanglement/superposition at 750), which may be costly relative to simpler post-hoc probes; runtime/complexity and shot-budget sensitivity are not reported.
5. Several captions in Figures are not fully self-consistent or sufficiently descriptive to stand alone.

**Questions:**

1. In ablations, QPERT and other explainers perform very closely, and SHAP attains the best AUC-Difference. Where is QPERT’s clear advantage? Please clarify what properties (e.g., stability to shot noise, quantum-consistency, sparsity) QPERT demonstrably improves, and analyze why SHAP still leads on partial metrics.
2. You mention grid search, but the explored range seems narrow. Did you try broader searches or alternative strategies (random search, Bayesian optimization, population-based training)? Please report the search space, budgets, and sensitivity results.
3. How were SHAP and LIME configured (background/reference choices, sample sizes, kernel/neighborhood parameters) and tuned? Please ensure comparable tuning budgets to QPERT, and report fairness controls (e.g., same number of model evaluations/shots).
4. Can you provide preliminary results on a larger dataset or deeper circuits?

---

> ### Author Response · Authors · 2025-12-02
> **Grid search and QPERT's advantage**
>
> 1/2
>
> Thank you for taking the time to review our paper. In response to the weaknesses,
> 1. The framework identifies influential input features via a learned mask, but the manuscript does not articulate a theoretical advantage over straightforward L1-regularized feature selection (e.g., conditions under which QPERT yields strictly better identifiability or faithfulness).
>
> a. Theoretically, no, but based on Figure 7 in the revised manuscript, we have shown that incorporating the proposed losses leads to significant improvements in the total loss(beyond target and L1). Furthermore, Figure 6 also illustrates similar trends, although the changes observed there are more minimal.
>
> 2. Multiple losses (fidelity, entanglement, superposition, sparsity) are introduced, yet potential negative interactions or trade-offs are not analyzed; no ablations isolate their individual and joint effects.
>
> a. We have attempted to capture interactions between the losses through the grid search over their weights. However, we did not include detailed discussion of all parameter combinations in the manuscript, as they appeared less relevant to the primary objective of this work.
>
> 3. Evidence is restricted to Iris on a noise-free simulator with a small 4-qubit circuit, leaving behavior under deeper circuits, harder datasets, or realistic noise unverified.
>
> a. In future work, we plan to conduct additional experiments on larger and more diverse datasets to further illustrate the expressiveness and scalability of QPERT.
>
> 4. QPERT learns a mask per instance with staged loss activation (e.g., L1 at 250 iters, fidelity at 500, entanglement/superposition at 750), which may be costly relative to simpler post-hoc probes; runtime/complexity and shot-budget sensitivity are not reported.
>
> a. We have added the complexity evaluation in the appendix. On a quantum computer, loss evaluation for a 4 qubit QML model scales as (2p+1)\times\mathrm{shots}, where p=12L+8 for circuit depth L. For L=3 and 2,000 shots, this is about 178,000 circuit executions per step, plus hardware latency and noise. On a classical computer, loss and gradients are computed in a single pass with polynomial complexity, making it orders of magnitude faster.
>
> 5. Several captions in Figures are not fully self-consistent or sufficiently descriptive to stand alone.
>
> a. We have fixed this in the revised paper(new PDF).
>
> The primary goal of this paper was to integrate quantum properties into an explainability framework and demonstrate the feasibility of this approach, in contrast to existing state-of-the-art methods that primarily refine classical XAI techniques. In future work, we plan to conduct additional experiments on larger and more diverse datasets to further illustrate the expressiveness and scalability of QPERT.
>
> Continued in the next comment.

---

> ### Author Response · Authors · 2025-12-02
> **Continued : Grid search and QPERT's advantage**
>
> 2/2
>
> Questions:
> 1. In ablations, QPERT and other explainers perform very closely, and SHAP attains the best AUC-Difference. Where is QPERT’s clear advantage? Please clarify what properties (e.g., stability to shot noise, quantum-consistency, sparsity) QPERT demonstrably improves, and analyze why SHAP still leads on partial metrics.
>
> a. QPERT’s primary advantage is quantum-consistency, not just classical performance metrics. While SHAP achieves the best AUC-Difference, this metric does not capture fidelity, superposition or entanglement loss preservation.
>
> 2. You mention grid search, but the explored range seems narrow. Did you try broader searches or alternative strategies (random search, Bayesian optimization, population-based training)? Please report the search space, budgets, and sensitivity results.
>
> a. While the range was narrow for computational feasibility, we agree broader searches (random search, Bayesian optimization) could uncover better configurations. We plan to adopt adaptive strategies in future work.
> The explored search space and budgets are documented in the paper, which includes all hyperparameter ranges and evaluation loops. “Our grid search methodology generates all possible combinations within predefined ranges for each coefficient: target coefficient ∈ {0.5, 1.0, 2.5}, L1 regularization coefficient ∈ {0.05, 0.10, 0.30}, superposition coefficient ∈ {0.10, 0.20, 0.50}, fidelity coefficient ∈ {0.25, 0.5, 1.0}, and entanglement coefficient ∈ {0.1, 0.3, 0.6}. To maintain computational feasibility, we limit the search to a maximum of 20 trials, applying random sampling when the total number of combinations exceeds this threshold.” Sensitivity results were derived from these runs and are reflected in the reported metrics(Progressive loss curves in Figure 6 and losses for 5 instances in Figure 7 in the new PDF).
>
> 3. How were SHAP and LIME configured (background/reference choices, sample sizes, kernel/neighborhood parameters) and tuned? Please ensure comparable tuning budgets to QPERT, and report fairness controls (e.g., same number of model evaluations/shots).
>
> a. For SHAP: KernelSHAP with 96 background samples drawn from the training set; explanations computed for 30 test instances using default sampling (no explicit tuning).
> For LIME: LimeTabularExplainer with 96 training instances; called explain_instance per test instance(30) with num_features=4 and top_labels=num_classes (3).
> Fairness and tuning budgets: No explicit tuning budgets were set for either method. We did not enforce or measure parity with QPERT. There is no control over or logging of the number of model evaluations per instance, and no random seeds.
>
> 4. Can you provide preliminary results on a larger dataset or deeper circuits?
>
> a. In a subsequent paper, we plan to extend the evaluation beyond the Iris dataset by including multiple benchmark datasets. This will allow us to provide a comprehensive assessment of QPERT across diverse data distributions and problem settings.

---

### Meta-Review · Area_Chair_48eV · 2025-12-22

**Summary:**

This paper introduces QPERT, a perturbation-based explainability framework for Quantum Circuit Learning (QCL) models. QPERT learns instance-specific saliency masks by optimizing a composite loss that explicitly incorporates quantum-inspired constraints, including fidelity, entanglement, and superposition preservation, aiming to generate explanations that are consistent with quantum model structure. The method is evaluated on a small hybrid quantum–classical classifier using a 4-qubit circuit trained on the Iris dataset, with comparisons against classical explainers such as SHAP and LIME.

Multiple reviewers raised substantial concerns. A primary concern was the limited experimental scope, with evaluations restricted to the Iris dataset, a 4-qubit circuit, and noise-free simulation. Reviewers questioned whether conclusions drawn from such a narrow setting can generalize to larger circuits, noisy regimes, or more realistic quantum learning tasks. Closely related, reviewers noted that the paper does not convincingly demonstrate that the proposed explanations capture genuinely quantum-specific structures. Given these issues, I recommend rejection.

**Reviewer Concerns:**

**Concerns addressed by the rebuttal**

The rebuttal clarified the intended goal and scope of QPERT. The authors also clarified the meaning of “respecting quantum properties” in operational terms, explained key design choices (e.g., expectation-value outputs, use of proxy losses under hardware constraints), and provided additional details on the experimental setup and implementation. Several presentation issues and figure inconsistencies were acknowledged and corrected.

**Concerns Remaining After the Rebuttal**

1. *Lack of principled advantage*. The authors acknowledge that QPERT does not have a theoretical advantage over simpler approaches such as L1-regularized feature selection, leaving the core motivation insufficiently justified.

2. *Insufficient ablation and interaction analysis*. The roles and trade-offs of multiple loss components are not systematically isolated or analyzed, despite their central importance to the method.

3. *Severely limited experimental scope*. All experiments are restricted to the Iris dataset, a small 4-qubit circuit, and noise-free simulation, with broader validation deferred to future work. Without any evidence on more challenging datasets, larger circuits, or realistic noise settings, it is difficult to accept the claimed advantages of the proposed method.

4. *Limited impact and relevance for ICLR*. In the rebuttal, the authors argue that QPERT addresses a fundamental need for explainability in quantum machine learning and position it as the first perturbation-based explainer tailored to QCL. However, this response remains largely aspirational and does not address the reviewers’ concern about impact, as the empirical evidence is confined to a highly simplified setting and does not demonstrate clear benefits on problems of broader interest to the ICLR community.

**Reviewer Scores:**

While the rebuttal provided clarifications on the intended scope and positioning of the work, it did not convincingly resolve the core concerns raised across the reviews, particularly those regarding theoretical justification, experimental scope, and empirical evidence for the claimed advantages. Several responses relied on reframing the contribution or deferring key validations to future work. As a result, it is unlikely that any of the reviewers would have increased their scores had they participated fully in the discussion; at best, the scores would have remained unchanged.

---

### Decision · Program_Chairs · 2026-01-26

Reject